

# Enhancing dance education through convolutional neural networks and blended learning

Zhiping Zhang[1] and Wei Wang[2]

[1] College of Education, HanJiang Normal University, Shiyan, Hubei, China
[2] Dancing College, Sichuan Normal University, Chengdu, SiChuan, China

## ABSTRACT

This article explores the evolving landscape of dance teaching, acknowledging the transformative impact of the internet and technology. With the emergence of online platforms, dance education is no longer confined to physical classrooms but can extend to virtual spaces, facilitating a more flexible and accessible learning experience. Blended learning, integrating traditional offline methods and online resources, offers a versatile approach that transcends geographical and temporal constraints. The article highlights the utilization of the dual-wing harmonium (DWH) multi-view metric learning (MVML) algorithm for facial emotion recognition, enhancing the assessment of students' emotional expression in dance performances. Moreover, the integration of motion capture technology with convolutional neural networks (CNNs) facilitates a precise analysis of students' dance movements, offering detailed feedback and recommendations for improvement. A holistic assessment of students' performance is attained by combining the evaluation of emotional expression with the analysis of dance movements. Experimental findings support the efficacy of this approach, demonstrating high recognition accuracy and offering valuable insights into the effectiveness of dance teaching. By embracing technological advancements, this method introduces novel ideas and methodologies for objective evaluation in dance education, paving the way for enhanced learning outcomes and pedagogical practices in the future.

## INTRODUCTION

Blended learning is a new form of teaching organization that adds online teaching modes to traditional teaching forms. There is no unified mode, but it combines the advantages of offline and online teaching to break through the time and space limitations of teaching activities and maximize the time and space conditions for teaching and learning. In practical operation, blended learning relies on rich online micro-lessons and video resources to teach basic knowledge and give students sufficient time for understanding and thinking. Then, offline, teachers provide targeted explanations of the key and difficult points and common problems students reflect in online learning, thus effectively improving students' learning efficiency and depth (*Yanfei, 2021*).

Corresponding author
Wei Wang, zzpgf913@163.com

Dance, as a form of artistic expression, is loved by many (*Fink et al., 2021*). People use dance to showcase their emotions, thoughts, and cultural backgrounds, express their personalities and styles, and convey information and values. Dance brings aesthetic pleasure and enjoyment to the audience and inspires emotional resonance and reflection, bringing joy and enlightenment to people's hearts (*Lee et al., 2019*). Additionally, dance can also exercise the body. Dance requires good physical and coordination abilities, which can help people exercise their muscles, improve their flexibility and endurance, and improve the health of their cardiovascular and respiratory systems. Dance can also enhance the coordination and balance of the body and mind, improve people's spatial and rhythmic sense, and help improve their physical fitness and athletic ability. In the field of education, dance also plays a significant role. Dance education can help students develop physical and cognitive skills, cultivate self-confidence and cooperation, and improve their cultural literacy and aesthetic ability. Dance education can also help students understand their own bodies and emotions, discover their potential and interests, and lay the foundation for their future development (*El Raheb et al., 2019*). Therefore, an objective evaluation system is needed to promote dance development.

The evaluation system for dance teaching is crucial to the teaching process. Its purpose is to assess students' performance in dance skills, movement expression, emotional expression, and other areas and provide corresponding guidance and feedback to promote students' growth and progress. Traditional dance teaching evaluation is mainly based on subjective evaluation by teachers, which has problems such as inconsistent evaluation standards, difficult-to-quantify evaluation results, and time-consuming evaluation processes. To address these issues, the use of multimodal data analysis to evaluate dance teaching courses has been proposed and applied.

In recent years, there has been a growing interest in applying advanced technologies to educational assessments to address these challenges. The intersection of dance education and technology presents a unique opportunity to enhance the accuracy and depth of performance evaluations. Specifically, the integration of convolutional neural networks (CNNs) with multimodal data analysis offers a promising approach to overcome the limitations of traditional methods. Dance is a complex art form that involves both technical skill and emotional expression. Capturing the full spectrum of a dancer's performance requires more than just visual observation; it necessitates a comprehensive analysis of both movement dynamics and emotional states. The motivation behind this study stems from the need to develop an evaluation framework that can objectively assess these multidimensional aspects of dance performances.The current research into dance teaching evaluation faces several significant gaps that need to be addressed to advance the field effectively. Traditional methods of dance assessment, heavily reliant on subjective teacher evaluations, suffer from inconsistencies in standards, difficulties in quantifying outcomes, and inefficiencies in processing. The shift toward multimodal data analysis presents a promising alternative; however, this approach itself is fraught with challenges that must be overcome. One critical gap is the integration and effective fusion of diverse data types, such as motion capture and audio signals, which is essential for a comprehensive assessment of both technical and emotional aspects of performance. While CNNs offer

automated feature extraction from high-dimensional data, they may not fully address the nuances of dynamic, multidimensional data inherent in dance performances. Moreover, the application of multi-view machine learning (MVML) algorithms, despite their potential for handling multi-view data, requires further exploration to validate their efficacy in practical settings. These algorithms' capability to map and classify complex emotional expressions remains underdeveloped, and their effectiveness in real-world scenarios, such as varying performance environments and individual differences, needs thorough examination. Addressing these research gaps by refining multimodal integration techniques and validating advanced machine learning algorithms in diverse contexts will be crucial for advancing objective and comprehensive dance teaching evaluations.

To sum up, traditional assessment methods, which rely heavily on subjective teacher evaluations, face issues such as inconsistencies, difficulties in quantifying outcomes, and inefficiencies. To address these challenges, this article aims to:

Develop a multimodal evaluation framework: This study introduces the MVML method based on the Dual Wing Harmonium (DWH) model to analyze students' facial expressions and dance movements simultaneously. By combining facial data for emotional expression and motion capture technology for dance motion, this framework provides a more comprehensive performance assessment.

Enhance data integration techniques: The article addresses the critical gap in integrating and effectively fusing diverse data types, such as motion capture and audio signals. This integration is essential for a holistic evaluation of dance performances' technical and emotional aspects.

Improve automated analysis with advanced machine learning: The study enhances automated feature extraction from high-dimensional data by utilizing improved CNNs for evaluating dance movements. This approach aims to capture dance performances' dynamic and multidimensional nature.

Validate multi-view machine learning (MVML) algorithms: The research explores the application of MVML algorithms in practical settings, focusing on their efficacy in mapping and classifying complex emotional expressions. This includes assessing their effectiveness across varying performance environments and individual differences.

## RELATED WORKS

In using multimodal data analysis to evaluate dance teaching, it is necessary to integrate and analyze multiple data types. For example, in motion capture, the student's motion data and audio signals must be fused to evaluate the student's motion and emotional expression comprehensively. Additionally, the student's psychological and behavioral characteristics need to be fused in emotion analysis to obtain more comprehensive and accurate emotional information. To achieve data fusion and analysis, various machine learning and deep learning algorithms need to be used.

The CNN algorithms can automatically extract features from high-dimensional data, such as images and audio, without requiring manual feature design (*Liu & Zhang, 2021*; *Hong & Yingzi, 2020*; *Mathis et al., 2020*). CNN algorithms reduce the number of

parameters that need to be trained, the complexity of the model, and the training time through weight sharing and local connections (*Dhiman & Vishwakarma, 2019*; *Singh & Vishwakarma, 2019*; *Qianwen, 2021*). In addition, CNNs have translation and local invariance, meaning that the feature extraction results for the same object in different positions and angles should be the same or similar, making CNNs robust and generalizable (*Gnana Priya & Arulselvi, 2019*; *Ho, Shim & Wee, 2020*; *Wu, Wang & Hu, 2020*). Therefore, by combining CNNs with motion capture data, the data collected from students' bodies can be processed, and objective evaluations can be made by analyzing the data and comparing it to excellent examples.

Additionally, in evaluating emotional expression, the MVML algorithm based on Dual Wing Harmonium (DWH) can be used (*Li et al., 2021*; *Shi et al., 2020*; *Han et al., 2021*). In traditional single-view classification tasks, feature extraction and classifier learning are typically used to complete the classification. However, for multi-view data, single-view feature extraction and classifier learning may not fully utilize the relationships between multi-view data, hence the need for the MVML algorithm for joint learning. The core idea of the MVML algorithm is to learn a metric matrix, which maps the data from multiple views to the same space for distance metrics, thus improving classification accuracy (*Chen et al., 2020*; *Yu et al., 2022*; *Qiu et al., 2022*; *Pang et al., 2021*; *Wang et al., 2022*). Specifically, the algorithm learns a discriminative metric matrix by minimizing the distance between the same object in different views and maximizing the distance between different objects in different views (*Juneja & Rana, 2021*; *Ali et al., 2021*; *Andrejevic & Selwyn, 2020*; *Wang et al., 2023*). This way, data from other views can be mapped to the same space, making it convenient for distance metric and classification tasks. Therefore, emotional expression evaluation can be solved.

Current research in dance evaluation systems has explored various methods for integrating technology into the assessment process. For example, motion capture technology has been employed to analyze the technical precision of dance movements, while machine learning algorithms have been used to interpret and evaluate these movements objectively. Additionally, audio-visual data fusion techniques have enabled a more holistic understanding of a dancer's performance, incorporating physical and emotional aspects. Moreover, recent studies have highlighted the importance of real-time feedback systems in dance education. These systems use sensors and algorithms to provide immediate feedback to students, helping them correct their movements and improve their performance more effectively than traditional methods. Such technological advancements enhance the accuracy of dance evaluations and contribute to more personalized and effective dance instruction.

## METHODS

In practical applications, to achieve a more scientific evaluation of students' dance proficiency and emotional expression, this article uses deep learning combined with motion capture technology and micro-expression recognition technology to optimize students' dance movements and emotional expression. The specific implementation

process is as follows: firstly, the DWH framework is used to preprocess video data and extract facial muscle movement information; then, graph convolutional networks and temporal convolutional networks are used to analyze micro-expressions of facial muscle images; at the same time, dance movements captured by infrared cameras and RGB cameras are processed and evaluated using CNN and finally, the student's dance movements and emotional expression are jointly analyzed. This article aims to help teachers have a more comprehensive understanding of students' performance and responses and improve the scientific accuracy of teaching evaluations through this method.

## Data extraction

To ensure student facial data collection completeness and protect student privacy, we need to preprocess their facial data. An MVML algorithm based on the DWH model is proposed to meet the emotional expression needs in dance teaching. The algorithm attempts to embed multiple views into a single low-dimensional latent variable space by extracting different information from multimodal data, minimizing the distance between similar data pairs, maximizing the distance between dissimilar data pairs, and learning the most appropriate distance metric in the subsequent supervised learning process. This article constructs the model based on pairwise constraints and optimizes it based on the Hinge loss function. Then, the model is initialized using random sampling.

Given a set of data $M = (x, z)$, a new function is obtained after mapping the data:

$$T = E_{p(h|x,z;\Psi)}[h] \tag{1}$$

where T represents the latent representation of embedding the data points $M = (x, z)$ in a shared low-dimensional latent variable space, while h represents hidden unit nodes, which can be regarded as observation results obtained from different sources for a given dataset by dividing it into similar or dissimilar two labels, we hope that the results obtained through metric learning are that the distance between the collection of similar data pairs in the mapping space is close, while the collection of dissimilar data pairs is far, to achieve better classification and clustering and assist teachers in the evaluation of course standards. This article uses the Euclidean distance to measure the distance between data points in the new mapping space. Since all data sets have the same representation and are denoted as $(y^{(i)}, y^{(j)})$ in this article, we define $S = \{y^{(i)}, y^{(j)}\}$ as the set of similar data pairs and $D = \{y^{(i)}, y^{(j)}\}'$ as the set of dissimilar data pairs, and the following optimization problem arises:

$$\min_{\Psi} \sum_{y^{(i)}, y^{(j)} \in D} \left\| t^{(i)} - t^{(j)} \right\|^2$$
$$s.t. \forall y^{(i)}, y^{(j)} \in D, \left\| t^{(i)} - t^{(j)} \right\|^2 \geq 1. \tag{2}$$

In the equation, y represents all the data that appears in S and D; $t^{(i)}$ is the representation of $y^{(i)}$ in the low-dimensional latent variable space. This optimization problem aims to make the distance between similar data pairs in the mapping space as close as possible and keep the distance between dissimilar data pairs as far as possible, as shown in Fig. 1. The low-dimensional latent variable space range is defined as a constant of 1. In this way, all the

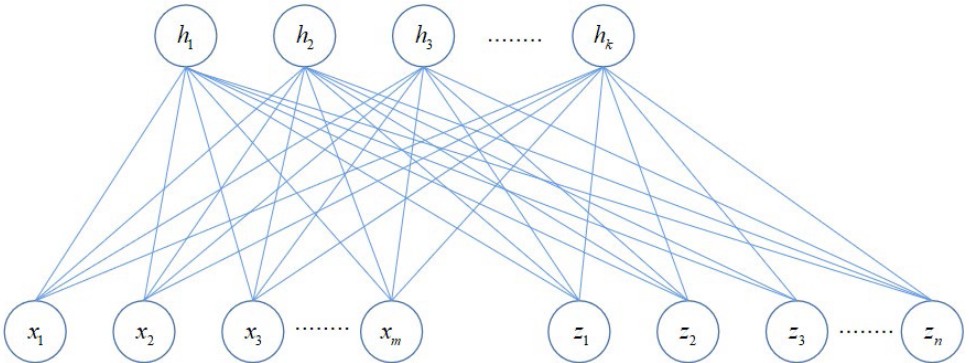

**Figure 1 The topological structure of DWH model.**

parameters that need to be learned in the model are only $\Psi$, which is already embedded in $t$.

We provide excellent standard emotional expression data through continuous observation and increasing the amount of observed data. Based on this data, we use the unsupervised DWH model in the framework of the supervised algorithm MVDML to learn the parameter $\Psi$ through maximum likelihood estimation, which promotes the comparison of similarity measures between different data point sets. Combining distance metric learning and maximum likelihood estimation, the optimization problem is rewritten in the following form:

$$\min_\Psi \frac{1}{|Y|}L(Y;\Psi) + \lambda\frac{1}{|S|}\sum_{(y^{(i)},y^{(j)})\in S}\left\|t^{(i)}-t^{(j)}\right\|^2$$
$$s.t. \forall(y^{(i)},y^{(j)})\in D, \left\|t^{(i)}-t^{(j)}\right\|^2 \geq 1 \tag{3}$$

$L(Y;\Psi)$ is the negative log-likelihood value of data Y, where Y is determined based on the parameter $\Psi$, $\lambda$ is the weight parameter, and $|\cdot|$ represents the cardinality of the set.

To solve the constraint problem presented in the optimization problem, a loss function can be used to eliminate the constraints. The Hinge loss can be used to eliminate the constraints of the optimization problem, and under the hinge loss, an unconstrained optimization problem can be obtained. Therefore, the subgradient algorithm can be used to optimize the problem. Under the Hinge loss, we can obtain a new optimization problem:

$$\min_\Psi \frac{1}{|Y|}L(Y;\Psi) + \lambda_1\frac{1}{|S|}\sum_{(y^{(i)},y^{(j)})\in S}\left\|t^{(i)}-t^{(j)}\right\|^2$$
$$+\lambda_1\frac{1}{|D|}\sum_{y^{(i)},y^{(j)}\in D}\max(0, 1-\left\|t^{(i)}-t^{(j)}\right\|^2) \tag{4}$$

Here, $\lambda_1$ and $\lambda_2$ are weighting parameters. The Hinge loss in Eq. (4) is 0 when the optimization problem satisfies the constraints. When the optimization problem in Eq. (3) does not satisfy the constraints, the Hinge loss is not 0.

## Contrastive divergence

To make the solution of the optimization problem more appropriate, the Hinge loss should be minimized, thus forcing the constraints to be satisfied. Since the Hinge loss is non-differentiable, this article uses the subgradient algorithm for optimization. Therefore, the contrastive divergence (CD) algorithm is adopted to roughly estimate the gradient of $\Psi$, which corresponds to the negative log-likelihood value $\frac{1}{Y}L(Y;\Psi)$. It is very easy to derive the gradient (or subgradient) based on the distance metric loss caused by similar data pairs and dissimilar data pairs here, so the subgradient solution formula is:

$$
\begin{aligned}
\nabla\theta_i &= E_p[\phi(x_i)] - E_{\hat{p}}[\phi(x_i)] \\
\nabla\eta_j &= E_p[\varphi(z_j)] - E_{\hat{p}}[\varphi(z_j)] \\
\nabla\lambda_k &= E_p[\chi(h_k)] - E_{\hat{p}}[\chi(h_k)]
\end{aligned}
\tag{5}
$$

$$
\begin{aligned}
\nabla W_i^k &= E_p[\phi(x_i)\chi(h_k)^T] - E_{\hat{p}}[\phi(x_i)\chi(h_k)^T] \\
&+ \lambda_1\frac{2}{|S|}\sum_{(y^{(m)},y^{(n)})\in S}(t_k^{(m)} - t_k^{(n)})(\frac{\partial t_k^{(m)}}{\partial W_i^k} - \frac{\partial t_k^{(n)}}{\partial W_i^k}) \\
&+ \lambda_2\frac{2}{|D|}\sum_{(y^{(m)},y^{(n)})\in D}(\left\|t_k^{(m)} - t_k^{(n)}\right\|^2 < 1)(t_k^{(n)} - t_k^{(m)})(\frac{\partial t_k^{(m)}}{\partial W_i^k} - \frac{\partial t_k^{(n)}}{\partial W_i^k})
\end{aligned}
\tag{6}
$$

$$
\begin{aligned}
\nabla U_j^k &= E_p[\varphi(z_j)\chi(h_k)^T] - E_{\hat{p}}[\varphi(z_j)\chi(h_k)^T] \\
&+ \lambda_1\frac{2}{|S|}\sum_{(y^{(m)},y^{(n)})\in S}(t_k^{(m)} - t_k^{(n)})(\frac{\partial t_k^{(m)}}{\partial U_j^k} - \frac{\partial t_k^{(n)}}{\partial U_j^k}) \\
&+ \lambda_2\frac{2}{|D|}\sum_{(y^{(m)},y^{(n)})\in D}(\left\|t_k^{(m)} - t_k^{(n)}\right\|^2 < 1)(t_k^{(n)} - t_k^{(m)})(\frac{\partial t_k^{(m)}}{\partial U_j^k} - \frac{\partial t_k^{(n)}}{\partial U_j^k})
\end{aligned}
\tag{7}
$$

where $E_p[\cdot]$ is the expectation with respect to the true distribution, $E_{\hat{p}}[\cdot]$ is the expectation with respect to the empirical distribution. However, the exact calculation of $E_p[\cdot]$ with respect to the true distribution is very difficult, so it is necessary to start with the expectation with respect to the empirical distribution and use Gibbs Sampling to approximate $E_{\hat{p}}[\cdot]$. The sampling process can be completed through iterative steps, as follows, where $l$ is the number of iterations for the cycle.

$$
\begin{aligned}
E[h_k^l] &= E_{p(h_k|x,z)}[h_k|E[x^{l-1}], E[z^{l-1}]] \\
E[x_i^l] &= E_{p(x_i|h)}[x_i|E[h^{l-1}]] \\
E[z_j^l] &= E_{p(z_i|h)}[z_j|E[h^{l-1}]].
\end{aligned}
\tag{8}
$$

## MVML algorithm

When introducing MVML algorithms in dance teaching, collecting information on students' movements and facial expressions becomes particularly important. Figure 2 shows the devices and their arrangements that we used in the information capture process. The blue square outlined in the middle is the information capture area where students stand to dance. Precise recording and capture of students' dance movements and facial expressions are possible through the seven yellow infrared cameras and four green RGB cameras mounted on the walls around the capture area, as shown in Fig. 2. The yellow

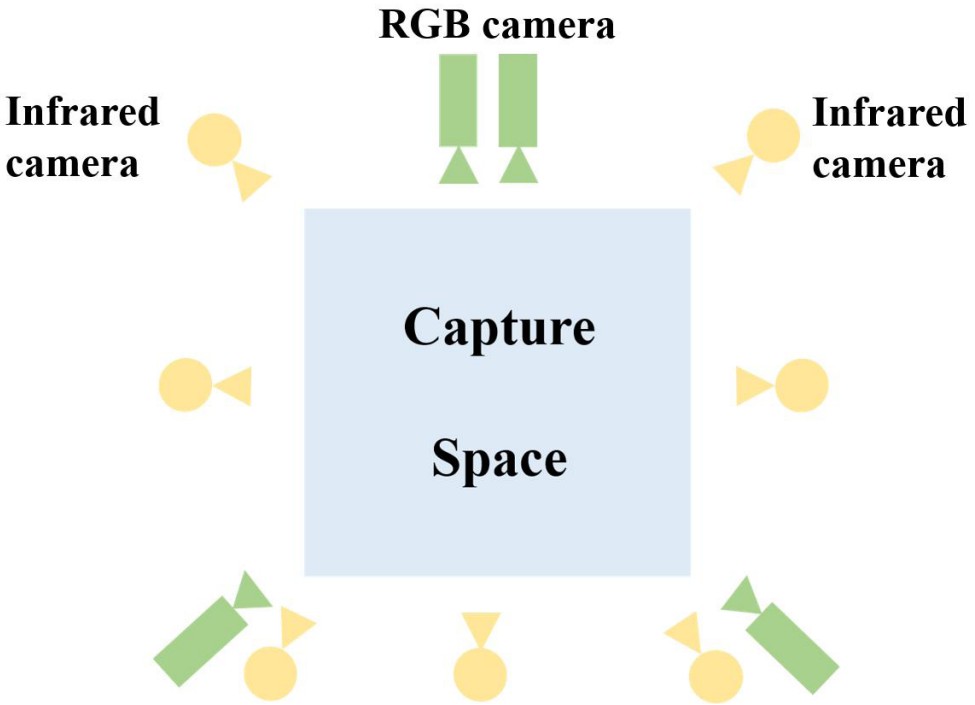

**Figure 2** **The layout of capture space.**

infrared cameras are used to capture ground truth posture data, and they are equipped with a motion capture (MOCAP) system to capture data through motion analysis. The green red-green-blue (RGB) cameras are used to record multi-view data. When the devices are activated, the images are captured from one viewpoint, and the data is quickly captured. When connected to the same hub, the cameras will automatically synchronize. Through this method, we can collect data on changes in students' movement and expression during their dance performances.

We preprocessed the motion capture data we collected to facilitate the subsequent processing by CNN. One method of preprocessing is to standardize the data. If the data is X, with a mean of μ and a standard deviation of $\sigma$, then the formula for standardizing the data can be represented as:

$$X_{std} = \frac{X - \mu}{\sigma}.$$ (9)

In the Eq. 9, $X_{std}$ represents the normalized data.

The collected data is input into a CNN, and after processing by the convolutional and pooling layers, image features are extracted. The input data is represented as $X_{std}$, where $X_{std}(i)$ represents the data collected at the i-th frame. After the convolution operation, the resulting feature map data is represented as $C_i$ as follows:

$$C_i = f[W * X_{std}(i) + b]$$ (10)

where W is the convolution kernel, b is the bias, and f is the activation function. The convolution operation can extract local features of the image, preserving the correlation between adjacent pixels. The pooling operation further reduces dimensionality and improves computational efficiency. The pooling formula is:

$$M = g(pool(C_i)) \tag{11}$$

where "pool" represents the pooling operation, and "g" is the activation function. After convolution and pooling operations, the feature map is input into the fully connected layer:

$$F = f(W_c * C + b_c). \tag{12}$$

In the equation, $W_c$ represents the weights of the output layer and $b_c$ represents the bias. Finally, classification or regression tasks are performed through the output layer:

$$Y = f(W_F * F + b_F) \tag{13}$$

where $W_F$ represents the weight of the output layer, and $b_F$ represents the bias. Finally, the output Y represents the evaluation result of the action, which can be used for classification or regression tasks.

## Evaluation process

We designed a quantifiable course evaluation system during the specific evaluation process of dance movements and emotional expressions. Based on the scores of each evaluation item, we classify students' levels in this module into three categories: poor, good, and excellent, to promote more hierarchical and targeted dance teaching activities. In our course evaluation system, a scoring system is used, which measures the standardization of dance movements and the accuracy of emotional expressions.

In the experiment, professional dance performers provided the standard data for scoring. As dance movements play a dominant role in the entire performance, we allocate weights to the dance movements and emotional expression modules in a ratio of 6:4 separately during scoring. By dividing the whole dance into several sections, we compare and analyze the student's movements and expressions with the professional dancer's data in each section, which obtains the initial scores from the system. Then, we add up the scores of the student's movement skills and emotional expression in each section and distribute them with a weighting ratio of 6:4. The final result is the total score. Assuming a dance can be divided into n sections, the score of movement skills in each section is $D_1; D_2 \dots D_n$, and the score of emotional expression in each section is $Q_1; Q_2 \dots Q_n$, then the total score can be described by the following formula:

$$S(Score) = 6/10(D_1 + D_2 + \dots + D_n) + 4/10(Q_1 + Q_2 + \dots + Q_n). \tag{14}$$

# EXPERIMENTS AND ANALYSIS

## Data collection and preprocessing

The preprocessing pipeline for analyzing facial expressions and motion capture data involved a series of meticulously executed steps to ensure accuracy and consistency in the data. For facial expression analysis, high-resolution camera feeds were initially captured from multiple angles to provide comprehensive coverage of the dancer's face. Image preprocessing was performed to standardize the input data, which included resizing images to a consistent resolution and normalizing them to adjust for varying lighting conditions. Techniques such as brightness and contrast adjustment were applied to mitigate the impact of diverse lighting environments.

Face detection was accomplished using a pre-trained deep learning model. This model was fine-tuned with a diverse dataset to handle variations in facial expressions and orientations effectively. Following face detection, facial landmark detection was carried out using algorithms like Dlib or MediaPipe to identify key facial features including the eyes, nose, and mouth. The detected landmarks were aligned with a standardized facial model to correct for head tilts and rotations using affine transformations. Data augmentation techniques, including rotation, scaling, and shifting, were employed to enhance the model's robustness and generalizability.

For expression recognition, a CNN was trained using a labeled dataset of facial expressions, encompassing emotions such as happiness, anger, sadness, and joy. The dataset was carefully prepared by filtering out mislabelled and inconsistent images, ensuring high-quality and representative samples. The CNN was trained incorporating dropout, batch normalization, and learning rate scheduling techniques to optimize performance. Validation and test sets were utilized to assess the model's accuracy and to prevent overfitting.

In the motion capture pipeline, silhouette extraction was achieved through background subtraction techniques. Gaussian Mixture Model (GMM) were employed to isolate the dancer from the background, followed by segmentation using morphological operations to refine the silhouette and eliminate noise. Post-processing steps included contour smoothing and hole-filling to ensure an accurate representation of the dancer's outline.

Pose estimation was performed using algorithms like OpenPose to detect key body joints and poses, such as shoulders, elbows, knees, and ankles. Calibration was necessary to account for variations in camera perspectives and dancer movements, ensuring consistent keypoint mapping across frames and cameras. Table 1 outliers and erroneous detections were filtered out using statistical methods and consistency checks to maintain accuracy in pose estimation.

Movement analysis involved extracting movement trajectories by tracking key joint positions over time, with techniques such as Kalman filters or particle filters applied to smooth the data and handle noise. Angles and movement parameters were calculated to evaluate the correctness of dance movements, involving computations of angles between body segments and analysis of movement patterns. Data normalization was performed to

**Table 1 Settings of parameter.**

| Component | Parameter | Value |
|---|---|---|
| DWH model | Latent variable space range | 1 |
| | Distance metric | Euclidean distance |
| | Optimization method | Hinge loss + Subgradient algorithm |
| | Weight parameter ($\alpha$) | 0.5 |
| | Regularization parameter ($\lambda$) | 0.01 |
| CNN | Batch size | 32 |
| | Learning rate | 0.001 |
| | Convolution kernel size | $3 \times 3$ |
| | Pooling type | Max pooling |
| | Activation function | ReLU |
| | Weight decay | 0.0005 |
| Contrastive divergence | Number of iterations | 1000 |
| | Gradient estimation method | Empirical distribution + Gibbs sampling |
| MVML model | Distance metric | Euclidean distance |
| | Optimization method | Hinge loss + Subgradient algorithm |
| | Weight parameter ($\beta$) | 0.5 |
| | Regularization parameter ($\lambda$) | 0.01 |
| | Weight ratio (Dance movements) | 6:4 |

address variations in dance styles and individual performances, standardizing trajectories and angles for consistent analysis.

This article uses the CPU of Xeon(R) E5-2640 v4, the GPU of 4*Nvidia Tesla V100 and the Ubuntu system to complete the environment setup and model training. The deep learning framework is Tensorflow. We set the learning rate of the model to 0.004, the training epoch to 40 rounds, the weight decay term to 0.001, and adopted MAE as the optimizer.

## Data source

This article emphasizes the importance of emotional expression in dance teaching. Dance performance is a visual art based on emotions, and even the most excellent dance works will lose their impact without proper emotional expression. The beauty of dance comes from human beings themselves, and our emotions fluctuate due to external influences. Dancers can effectively communicate their intrinsic values by synthesizing emotional expression with choreographed movements. Humans exhibit a spectrum of affective states: happiness, anger, surprise, fear, disgust, sadness, and neutrality. Each emotion corresponds to distinct facial expressions, which can be accurately identified through advanced video analysis techniques.

We used DanceDataset (https://zenodo.org/records/5118689, DOI: 10.5281/zenodo.5118689). Cultural sensitivity is crucial in the study of emotional expression in dance. Emotions and their expression can be culturally specific, so we focus on these cultural differences to avoid misinterpreting or exploiting traditional dance practices. Cultural context should be respected, recognizing that conventional dance forms may embody

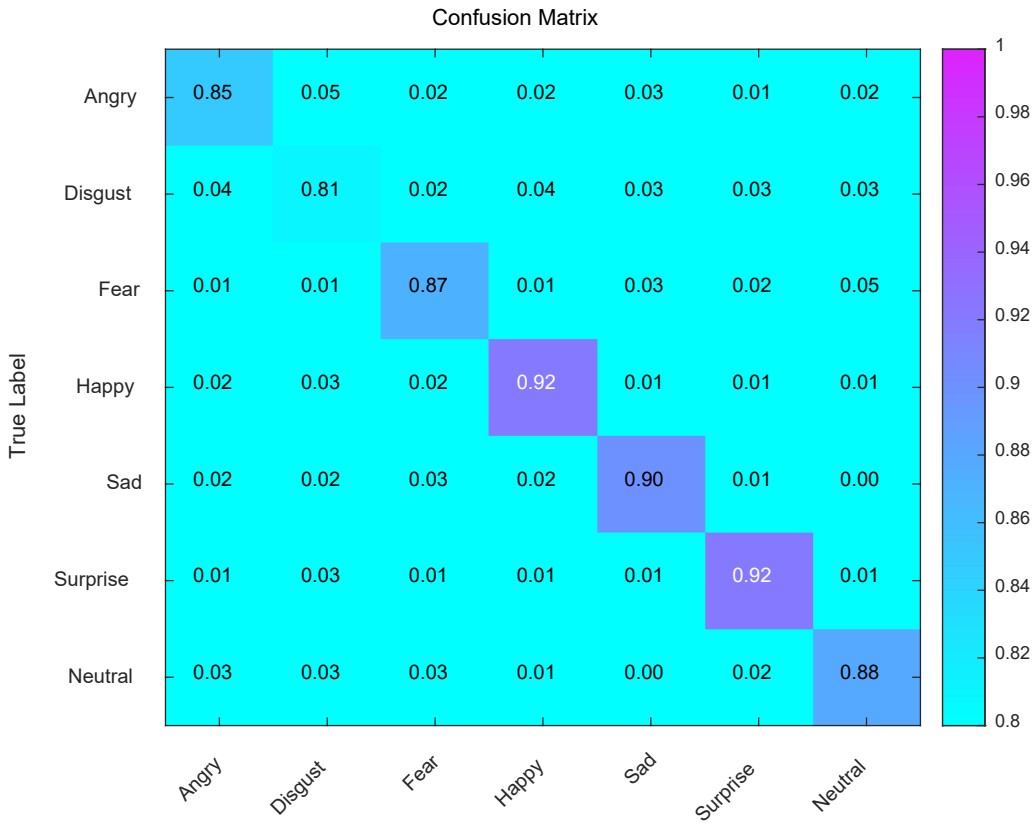

**Figure 3** Confusion matrix analysis of expression recognition.

unique emotional expressions inherent in their cultural heritage. It is important to approach these practices with sensitivity and respect, ensuring that cultural traditions are not distorted or inappropriately appropriated. Addressing potential bias in sentiment analysis models is another important ethical issue. Acknowledge bias in the data and remove it by ensuring that the dance dataset is diverse and representative of various demographic and cultural groups. This approach prevents the continuation of bias in the analysis and application of sentiment data. Furthermore, fairness in the application ensures that all dancers, regardless of their background, have an equal opportunity to participate in and benefit from the analysis.

## Performance of expression recognition

Using the DWH model, the confusion matrix was constructed to display the recognition accuracy of different facial expressions. As shown in Fig. 3, the recognition accuracy of facial expressions is generally above 80%, and it can further improve with continuous training. The highest recognition accuracy is achieved for the happy and surprised expressions, with an accuracy rate of 92% and a probability of being incorrectly detected of only 8% for other expressions. Happiness and surprise are always displayed with an uncontrollable uplift of the corners of the mouth and widened eye pupils, whether in real life or during

**Table 2 Results of model comparsion.**

| Model | Precision | Recall | F-score |
|---|---|---|---|
| GCNN | 0.8975 | 0.9011 | 0.8992 |
| RGAN | 0.9094 | 0.9031 | 0.9062 |
| SWHIM | 0.9392 | 0.9507 | 0.9449 |
| DWH-MVML | 0.9825 | 0.9802 | 0.9813 |

a dance performance. The muscles in our faces reflect our inner emotions. Therefore, recognizing happy and surprised expressions is relatively easier and more accurate. Disgust expression appears less frequently in dance performances because dance is an extreme expression of love for life. Therefore, the recognition accuracy for disgust expression is 81%. The recognition accuracy for angry, fearful, and neutral expressions is 85%, 87%, and 88%, respectively, and none of them have reached 90%, indicating a need for further training. Sadness expression is a common emotion, and its recognition accuracy is 90%. With continuous training, the recognition accuracy of facial expressions will continue to improve. By recognizing facial expressions, teachers can make scientific judgments, which not only helps improve student performance but also reduces the pressure on the teacher and increases work efficiency.

## Model comparison

This article employs the F-score as the evaluation criterion for the model. Table 2 compares the model's performance after data preprocessing and normalization with several other models.

In comparison to the graph convolutional neural network (GCNN), regularized generative adversarial network (RGAN), and stochastic Wasserstein histogram matching (SWHIM), the Deep Weighted Hybrid Multi-View Multi-Label (DWH-MVML) algorithm exhibits a significant enhancement in performance. Specifically, DWH-MVML enhances precision, recall, and F-score by 9.46%, 8.77%, 9.13% over GCNN, and 8.03%, 8.53%, and 8.29% over RGAN. When compared to SWHIM, DWH-MVML achieves increases of 4.61%, 3.10%, and 3.85% in precision, recall, and F-score, respectively. These improvements underscore DWH-MVML's superior handling of complex prediction tasks, significantly boosting model accuracy and stability and highlighting its leading position in the current research landscape.

## Learning effect evaluation

To explore the changes in student performance after introducing MVML for course evaluation, we selected twenty students as our subjects. We compared their dance learning results before and after introducing this method to evaluate its effectiveness. As shown in Fig. 4, we observed that before the introduction of MVML, the expression of student emotions followed a normal distribution trend according to the teacher's grading standards, ranging from the 60s to the 90s with a large span. Moreover, many students were hovering around the passing mark, not because of the teacher's insufficient level but because emotional expression is subtle and requires total attention throughout the dance

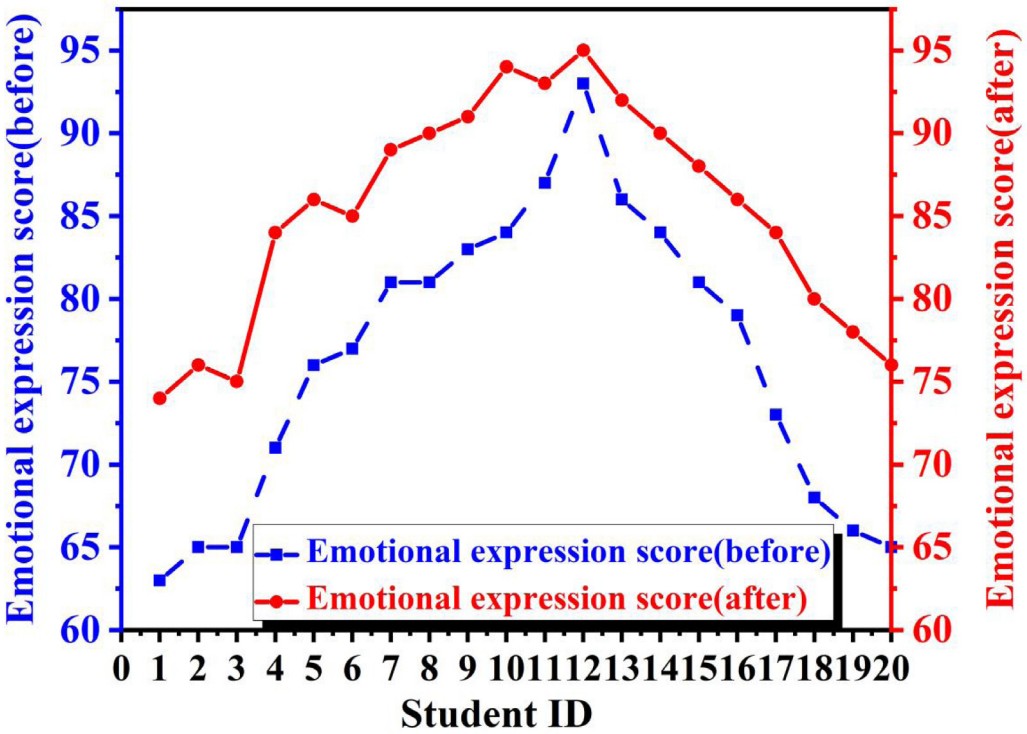

**Figure 4   Emotional expression ability.**

performance. Given limited time and the context of mixed education with more students, many students could only receive guidance and evaluation online, and the teacher could only give a rough assessment based on their experience, often overlooking many details and causing slow progress for the students. However, through the MVML algorithm, objective evaluations can be conducted from multiple dimensions, and with the help of big data, objective evaluations can be given by identifying student emotions and providing improvement suggestions.

After the introduction of MVML, we found that the evaluation system became more objective and comprehensive, and teachers could give more reasonable suggestions for different student issues. As a result, students made significant progress in emotional expression after the introduction, and even those who had barely passed previously were now performing well in this area. Even outstanding students made progress under scientific guidance. This is evident in the graph, which illustrates a substantial improvement. This demonstrates that our method is of great help to students, providing a more objective evaluation system. Guided by empirical data, students can continuously refine their skills by addressing their weaknesses, resulting in a marked enhancement in their ability to express emotions through dance.

As the most essential part of dance, the standardization of dance movements directly affects the level and visual experience of the entire dance performance. Dancers with standard and accurate dance postures and movements often possess more skilled dance

expression techniques, providing the audience with a better visual experience and sensory enjoyment. In practical dance instruction, ensuring that students' movements are standardized is a primary focus for teachers. Traditionally, teachers teach dance movements to students by demonstrating them, and students learn gradually by observing and imitating the teacher's movements. Following the teacher's demonstration, corrective guidance on students' movements and postures is provided through a one-to-many teaching approach. However, due to limited course time and human resources, it is not practical for teachers to provide detailed guidance on the dance movements of each student. Furthermore, when multiple decomposed movements are integrated and performed rapidly, the immediacy and coherence of the dance characteristics make it difficult to discern the detailed nuances of the individual movements. Due to the limitations of human eyesight and brain reaction speed, teachers' guidance to students has also been neglected in many areas. Given the challenges above, the implementation of an MVML evaluation system offers an effective solution for addressing these issues.

Here, we selected 20 samples from the dance training class as research samples. We captured and analyzed their dance movements using an MVML system and then generated feedback reports based on the evaluation results, guiding them to improve their movements, thereby promoting the standardization of their dance movements. In Fig. 5, we compared the standardization scores of these 20 students before and after the introduction of the MVML system to measure the system's effectiveness. Before the introduction of the MVML system (as shown by the blue dashed line in the figure), the standardization scores of these 20 students varied greatly, ranging from the 30s to the 60s, showing a great degree of difference and span. This was mainly to ensure the accuracy and credibility of the experimental results, and we followed the principle of randomness in sample selection. As the dance foundations of the students in the dance training class were not the same, it was only necessary to conduct personal longitudinal comparisons after the introduction of the MVML system to detect the system's effectiveness in improving learning outcomes.

The scores of each student after the introduction of the system are shown by the red dots in the figure. By comparing them with the corresponding blue squares, we found that the red dots were always above the blue squares, which means that the standardization of the students' dance movements had improved after introducing this evaluation system. In particular, students with previously weak foundational skills demonstrated substantial improvement after undergoing accurate movement recognition and assessment using the multi-view metric system. Through continuous training of body coordination based on feedback from the learning system and enhancement of muscle memory, these students achieved notable advancements in the standardization of their dance movements. On the other hand, students who had previously had a good foundation in dance showed even better standardization after being guided by the MVML system, with some students even approaching a professional level. These experimental results indicate that the MVML system significantly promotes the standardization of students' dance movements. Leveraging the power of emerging technologies greatly liberates limited human resources. It compensates

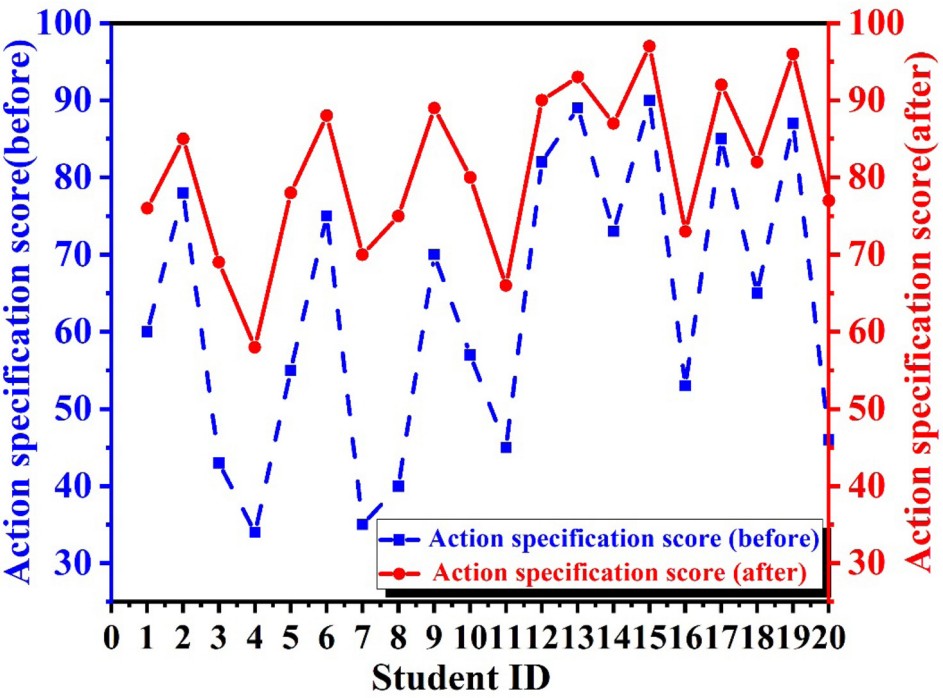

**Figure 5** Action specification ability.

for the deficiency of human eyes in recognizing subtle movements, paving the way for future dance teaching and performance.

## DISCUSSION

The study's findings are constrained by the relatively small sample size of twenty students drawn from a single institution. This sample size, while providing preliminary insights, restricts the ability to generalize the results to a broader population. Future research should address this limitation by incorporating larger and more diverse participant groups from various educational institutions and geographic locations. By increasing the sample size and diversity, more representative data can be collected, thereby improving the generalizability of the findings and offering a more comprehensive assessment of the system's effectiveness across various contexts. In addition, the study primarily focused on the technical performance of the MVML system without thoroughly exploring its integration into everyday teaching practices or its impact on user experience. The ease of integrating the system into existing teaching methodologies, user satisfaction and the associated learning curve were not thoroughly investigated.

The current study provides valuable short-term data but does not address the long-term sustainability of the observed improvements. While the results demonstrate improved emotional expression and dance performance in the short term, whether these enhancements persist over extended periods remains uncertain. Longitudinal studies are necessary to evaluate the durability of the impact of the MVML system. Tracking students'

progress over months or years will offer insights into whether the system's benefits are sustained and how continuous use influences students' development and performance. Moreover, the study does not account for contextual and cultural variations that may affect the effectiveness of the MVML system. Dance practices and emotional expressions can vary widely across different cultures and settings. Future research should explore how the system performs in diverse cultural contexts and educational systems. The generalizability of the MVML system to different educational environments and dance genres is another area that requires further exploration. The study was limited to a specific dance style and educational setting, so future research should test the system across a range of dance training programs, including classical, contemporary, and folk dance, as well as in different educational levels from primary to higher education.

Finally, the study did not address the cost and accessibility of implementing the MVML system on a larger scale. Future research should explore the financial and logistical aspects of scaling the system, including initial setup costs, maintenance, and training requirements. Evaluating the feasibility of widespread implementation will help determine how to make the system accessible to a broader range of educational institutions and users, addressing potential budget constraints and technical support needs.

## CONCLUSION

This article investigates the integration of dance instruction with emerging technologies within a hybrid environment through a systematic experimental design. It introduces an MVML system utilizing a bi-wing reed organ to capture students' facial data, enabling the recognition of various emotions and assessing whether students' emotional expressions are adequate during dance performances. Experimental trials involving twenty students demonstrated significant improvements in their ability to express emotions before and after implementing this system. Additionally, the scientific evaluation system provided teachers with more accurate feedback on students' emotional expression abilities, leading to corresponding improvements in their performance scores. Furthermore, the quality of students' dance performances is intricately connected to their body movements. By leveraging motion capture technology to gather data on students' body movements during performances and subsequently processing this data through CNNs, a more comprehensive and objective evaluation of the students' dance movements can be achieved. The experimental results show that, compared with the learning effect before the introduction of CNN and motion capture technology, the normativity of students' dance movements has greatly improved after the introduction of this system. Furthermore, by using facial expression and motion capture technology, comprehensive analysis can be performed under the support of CNN to generate feedback reports on dance emotional expression ability and movement normativity, providing guidance for students to improve their dance level and compensating for the limited human resources in reality, assisting teachers to complete teaching tasks faster and better.

Despite these advancements, several unresolved questions and limitations remain. First, the generalizability of these findings across diverse dance styles and age groups has not been

fully addressed. Future research should explore how the MVML system performs in different dance genres and with a broader range of participants. Additionally, while the current study demonstrates improvements in dance performance and emotional expression, the long-term impacts of these technologies on students' overall dance development and engagement remain unclear.

Another limitation is the reliance on a relatively small sample size, which may affect the robustness of the results. Future studies should include larger and more diverse participant groups to validate and extend these findings. Moreover, the integration of other advanced technologies, such as augmented reality or virtual reality, could further enhance the MVML system's capabilities and provide new insights into its effectiveness.

### Funding

This is supported by a general teaching and research topic of Hanjiang Normal University—"Research on the ideological and political implementation path of "Dance Foundation" under the mixed online and offline teaching mode", Project number 272022B19. The funders had no role in study design, data collection and analysis, decision to publish, or preparation of the manuscript.

### Grant Disclosures

The following grant information was disclosed by the authors:
Hanjiang Normal University; Dance Foundation: 272022B19.

### Competing Interests

The authors declare there are no competing interests.

### Author Contributions

- Zhiping Zhang conceived and designed the experiments, performed the experiments, analyzed the data, prepared figures and/or tables, authored or reviewed drafts of the article, and approved the final draft.
- Wei Wang conceived and designed the experiments, performed the experiments, performed the computation work, prepared figures and/or tables, authored or reviewed drafts of the article, and approved the final draft.

### Data Availability

The code is available in the Supplementary File.

The data is available at Zenodo: Manuel Cuadrado-García, Maja Seric, & Juan D. Montoro-Pons. (2021). DanceDataset (dataset) [Data set]. Zenodo. https://doi.org/10.5281/zenodo.5118689.

### Supplemental Information

Supplemental information for this article can be found online at http://dx.doi.org/10.7717/peerj-cs.2342#supplemental-information.

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
