# Peer review of "Enhancing dance education through convolutional neural networks and blended learning"

_PeerJ Computer Science, doi:10.7717/peerj-cs.2342_

## Round 0.1 · original submission · Major Revisions

Dear authors,

Thank you for the submission. The reviewers' comments are now available. It is not suggested that your article be published in its current format. We do, however, advise you to revise the paper in light of the reviewers' comments and concerns before resubmitting it.

Best wishes,

Reviewer 1 ·

Basic reporting

To improve the study on "Enhancing Dance Education through Convolutional Neural Networks and Blended Learning," I have offered several key suggestions:

The current study focuses on limited dance styles, which may not fully demonstrate the model's capabilities. Expanding the range to include various dance forms such as ballet, hip-hop, contemporary, and cultural dances will enhance the model's applicability and validity across different dance traditions.

The study uses basic motion capture techniques, which may not adequately capture the complexity of dance movements. Integrating more sophisticated motion capture and computer vision technologies, such as markerless motion capture systems, can improve the accuracy and detail of the captured movements, leading to more precise data for training the model.

Short-term studies may not fully capture the impact of AI-assisted dance education. Conducting long-term research will help evaluate the sustainable effects of the model on dance education. This includes assessing how the model influences students' learning progression, retention of dance skills, and long-term engagement with dance practice.

Experimental design

The study should include a comprehensive comparison between the MVML-DWH model and traditional dance teaching methods. This comparison should highlight the strengths and limitations of the AI-driven model, providing a clearer understanding of its advantages and potential areas for improvement. Including metrics such as learning outcomes, student engagement, and instructional efficiency will provide valuable insights.

The study should explore future research directions, including hybrid teaching models that combine AI with traditional instruction. Additionally, investigating the psychological impact of AI integration on students, such as changes in motivation, confidence, and learning anxiety, will provide a holistic view of the model's influence on dance education.

Validity of the findings

The current evaluation is limited to a specific academic context. To assess the model's generalizability, it is essential to test its practical applicability in diverse cultural and academic settings. This includes conducting studies in different countries, educational institutions, and among various age groups to determine how well the model adapts to different environments.

Transparency in research is vital for validation and replication. Making the data sources used in the study publicly available will enable other researchers to replicate the study, validate the results, and build upon the findings. This openness will enhance the credibility and impact of the research.

Additional comments

Grammar should be improved.

Reviewer 2 ·

Basic reporting

The manuscript attempts to explore the application of convolutional neural networks (CNNs) and blended learning in dance education. While the topic is novel, the manuscript falls short in several critical areas:Language and

Presentation:The manuscript lacks clarity and precision in its use of professional English. Numerous grammatical errors and awkward phrasing impede comprehension.The introduction fails to adequately contextualize the study within the existing body of literature. Key references are either missing or inadequately discussed.Structure and

Format:The structure of the manuscript does not conform to PeerJ standards. Specifically, the segmentation of sections is inconsistent and lacks logical flow.

Background and Motivation:The introduction does not sufficiently explain the motivation behind integrating CNNs into dance education. The rationale provided is superficial and does not engage with the core challenges or gaps in the field.

Literature Review:The literature review is not comprehensive. It lacks a critical examination of existing studies on the application of CNNs in educational settings and does not adequately justify the novelty of this study.

Suggestions for Improvement:A thorough revision of the manuscript's language and structure is necessary.The introduction and literature review sections need to be significantly expanded to provide a robust background and rationale for the study.

Experimental design

The experimental design is a crucial aspect of this manuscript, but several issues undermine its validity:

Scope and Aims:The article's content aligns with the aims and scope of the journal, but the study's objectives are not clearly articulated.

Methodology:The methods section is inadequately detailed. It does not provide sufficient information on the datasets used, the preprocessing steps undertaken, or the specific configurations of the CNN models.The description of the blended learning approach is vague and lacks concrete implementation details.

Ethical Standards:There is no discussion on the ethical considerations related to data collection and the involvement of human participants, which is a significant oversight.

Suggestions for Improvement:The methodology needs to be rewritten with detailed descriptions of all experimental steps, including data preprocessing, model configurations, and evaluation metrics.Include a comprehensive discussion on ethical considerations.

Validity of the findings

The validity of the findings is questionable due to several critical flaws:

Experimental Results:The results are presented without adequate statistical analysis. There is no discussion on the significance of the findings or the reliability of the results.The experimental setup does not seem to account for potential biases, and the evaluation metrics used are not justified.

Conclusions:The conclusions drawn are overly broad and not well supported by the data presented. The manuscript does not address the limitations of the study or suggest future research directions.

Suggestions for Improvement:Perform a rigorous statistical analysis of the results and include a discussion on the significance and reliability of the findings.Revise the conclusion to provide a balanced view of the study's outcomes, limitations, and potential future work.

Additional comments

The manuscript requires significant revisions to meet the journal's standards. The authors should consider seeking the assistance of a professional editor to improve the manuscript's language and presentation.A more detailed and systematic approach to the experimental design and analysis is essential for the manuscript to make a meaningful contribution to the field.

---

## Round 0.2 · Minor Revisions

Dear authors,

Thank you for submitting your revised manuscript. Feedback from the reviewers is now available. Although reviewer 1 is satisfied with the new version, based on the review 2, your article has not been recommended for publication in its current form. However, we encourage you to address the concerns and criticisms of reviewer 2 and to resubmit your article once you have updated it accordingly.

Best wishes,

**Language Note:** The review process has identified that the English language must be improved. PeerJ can provide language editing services - please contact us at [email protected] for pricing (be sure to provide your manuscript number and title). Alternatively, you should make your own arrangements to improve the language quality and provide details in your response letter. – PeerJ Staff

Reviewer 1 ·

Basic reporting

The authors incorporated all the given suggestions.

Experimental design

The authors incorporated all the given suggestions.

Validity of the findings

The authors incorporated all the given suggestions.

Additional comments

The authors incorporated all the given suggestions.

Reviewer 2 ·

Basic reporting

Basic Reporting:
Language and Presentation:

The authors have improved the language and presentation of the manuscript. However, some sections still contain grammatical errors and awkward phrasing that impede comprehension. Further proofreading by a professional editor is recommended.
The introduction now provides a better context for the study, but additional references and a more in-depth discussion of existing literature on CNNs in educational settings are needed to strengthen the background and rationale.
Structure and Format:

The manuscript's structure has been improved, but some sections still lack logical flow. Clearer segmentation of sections and a more consistent format are needed.
The literature review has been expanded but still lacks a critical examination of existing studies. A more comprehensive review that justifies the study's novelty is necessary.
Background and Motivation:

The introduction now better explains the motivation behind integrating CNNs into dance education. However, the rationale provided remains somewhat superficial. A deeper engagement with core challenges and gaps in the field is needed.
Suggestions for Improvement:

Thorough revision of the manuscript's language and structure.
Expansion of the introduction and literature review to provide a robust background and rationale for the study.

Experimental design

Scope and Aims:

The study's objectives are now more clearly articulated. However, further clarity is needed to align the aims with the experimental design.
Methodology:

The methodology section has been rewritten with more detailed descriptions of data collection, preprocessing, and model configurations. However, some steps still lack sufficient detail.
The description of the blended learning approach is more concrete, but further details on implementation are required.
Ethical considerations related to data collection and human participants have been added, but a more comprehensive discussion is needed.
Suggestions for Improvement:

Further detail in the methodology, particularly on data preprocessing, model configurations, and evaluation metrics.
A more comprehensive discussion on ethical considerations.

Validity of the findings

Experimental Results:

The results section now includes statistical analysis, but further discussion on the significance and reliability of the findings is needed.
The evaluation metrics are better justified, but potential biases in the experimental setup need to be addressed.
Conclusions:

The conclusions are more balanced and better supported by the data. However, the discussion on the study's limitations and future research directions needs to be expanded.
Suggestions for Improvement:

Perform a more rigorous statistical analysis of the results.
Revise the conclusion to provide a balanced view of the study's outcomes, limitations, and potential future work.

Additional comments

The manuscript has made significant strides in addressing the major concerns raised in the initial review. However, further refinements are necessary to meet the journal's standards.
The authors should consider seeking the assistance of a professional editor to improve the manuscript's language and presentation.
A more detailed and systematic approach to the experimental design and analysis is essential for the manuscript to make a meaningful contribution to the field.
Overall, the manuscript is much improved and on the right track, but additional work is required to ensure it meets the high standards of PeerJ Computer Science.

---

## Round 0.3 · Minor Revisions

Dear authors,

Thank you for your revised paper. Feedback from the Reviewer 2 is now available. We would encourage you to address the minor concerns and criticisms raised by Reviewer 2 and resubmit your article once you have updated it accordingly.

Best wishes,

Reviewer 2 ·

Basic reporting

The manuscript is generally well-written and uses clear, professional English throughout. The introduction effectively introduces the subject matter and provides a solid background on the integration of convolutional neural networks (CNNs) and blended learning in dance education. The literature is well-referenced, with relevant citations that support the discussion. However, I suggest clarifying the motivation behind the study more explicitly in the introduction to enhance the context for readers unfamiliar with the intersection of dance education and CNNs.

Experimental design

The experimental design is within the aims and scope of the journal and adheres to a high technical standard. The methods are described in sufficient detail, allowing for replication of the study. However, the discussion on data preprocessing could be expanded. Specifically, I recommend including more details on how the data was cleaned and prepared before being fed into the CNNs. Additionally, while the evaluation methods and assessment metrics are adequate, a brief discussion on the choice of model selection methods would improve the clarity and rigor of the study.

Validity of the findings

The findings are valid and supported by the experiments conducted. The arguments presented align well with the goals set out in the introduction. The conclusions are well-stated and are limited to the supporting results. However, the conclusion could benefit from a more explicit identification of unresolved questions, limitations, or suggestions for future research directions. This would provide a clearer roadmap for subsequent studies in this area.

Additional comments

Now Overall, the manuscript is a valuable contribution to the field of dance education and the application of CNNs in this context. The integration of technology in traditional education domains like dance is innovative and timely. I suggest minor revisions to address the points mentioned above, particularly in the areas of data preprocessing details, model selection rationale, and the expansion of future research directions in the conclusion.

---

## Round 0.4 · accepted · Accept

Dear authors,

Thank you for clearly addressing all the concerns and criticisms. It seems that all necessary additions and modifications are performed. Your paper is acceptable after this last revision.

Best wishes,

Reviewer 2 ·

Basic reporting

Accepted

Experimental design

Accepted

Validity of the findings

Accepted